# New Insights into Polymorphisms in Candidate Genes Associated with Incidence of Postparturient Endometritis in Ossimi Sheep (*Ovis aries*)

Fatmah A. Safhi [1] and Ahmed Ateya [2,*]

1 Department of Biology, College of Science, Princess Nourah bint Abdulrahman University, P.O. Box 84428, Riyadh 11671, Saudi Arabia; faalsafhi@pnu.edu.sa
2 Department of Development of Animal Wealth, Faculty of Veterinary Medicine, Mansoura University, Mansoura 35516, Egypt
* Correspondence: dr_ahmedismail@mans.edu.eg or ahmed_ismail888@yahoo.com; Tel.: +20-10-0354-1921; Fax: +20-502372592

**Abstract:** This study examined the genes related to immunity, metabolism, and antioxidants that may interact with the prevalence of postpartum endometritis in Ossimi sheep. We used fifty endometritis-positive Ossimi sheep and fifty that appeared to be normal. For the purpose of taking blood samples, each ewe had its jugular vein pierced. Nucleotide sequence differences for the immunological (alpha-2-macroglobulin, toll-like receptor 2, transforming growth factor beta, interleukin 1 receptor-associated kinase 3, high-mobility group box 1, Fc alpha and Mu receptor, and inducible nitric oxide synthase), metabolic (ADAM metallopeptidase with thrombospondin type 1 motif 20, potassium sodium-activated channel subfamily T member 2, Mitogen-activated protein kinase kinase kinase 4, FKBP prolyl isomerase 5, and relaxin family peptide receptor 1), and antioxidant (superoxide dismutase, catalase, NADH: ubiquinone oxidoreductase subunit s5, and Heme oxygenase-1) genes were found among sheep with endometritis and those in good condition utilizing PCR-DNA sequencing. Fisher's exact test revealed a significant difference in the probability of dispersal of all significant nucleotide changes between ewe groups with and without endometritis ($p < 0.01$). In endometritis ewes, there was a considerable up-regulation of the expression levels of *A2M*, *TLR2*, *IRAK3*, *HMGB1*, *FCAMR*, *iNOS*, *ADAMTS20*, *KCNT2*, *MAP3K4*, *FKBP5*, *RXFP1*, and *HMOX1*. Conversely, there was a down-regulation of the genes that encode *TGF-β*, *SOD*, *CAT*, and *NDUFS5*. The kind of marker and its frequency in postparturient endometrtits significantly impacted the transcript levels of the indicators under analysis. The results validate that nucleotide changes and gene manifestation outlines in these candidates are significant predictors of the prevalence of endometritis in sheep.

**Keywords:** Ossimi sheep; candidate genes; nucleotide variation; postparturient endometritis





## 1. Introduction

Sheep are regarded as vital farm animals as they play a significant role in sustaining human populations [1]. By increasing these animals' reproductive ability and production, this demand can be effectively supplied [2]. Ossimi is a prominent sheep breed in Egypt, which supplies around 6% of the nation's total production of red meat [3]. They are found across Egypt, particularly in the northern Nile Delta and along the western Mediterranean coast [4]. They are tiny to medium-sized with a fat tail and coarse wool. Adult body weight ranges from 51 to 53 kg, and the annual milk output is around 65 kg [5].

Endometritis was considered to be one of the mechanisms causing ewes to be subfertile [6]. Endometritis is the medical term for inflammation of the uterine endometrium. The majority of uterine inflammatory lesions are infectious in origin and are brought on by ascending infections that ordinarily block the lower genital canal or by infectious agents injected into the uterine cavity during mating, artificial insemination, or postpartum [7].

The clinical symptoms of endometritis include vaginal discharge that is thin, watery, possibly purulent, malodorous, associated with retained fetal membranes, dystocia, retained dead lambs, abortion [8]. Similar to cattle, endometritis in ewes most frequently occurs during the luteal phase or after delivery [8] and causes embryonic loss through either direct embryo cytolysis or rupture of uterine tissue. Additionally, Graafian follicle development and ovulation can be inhibited by the absorption of bacterial components [9]. Luminal epithelium regeneration in the ewe does not start until after Day 8 and is finished by Days 28 to 31 postpartum [10]. Therefore, the natural mucosal defense system is compromised for a considerable amount of time following lambing.

The amount and severity of the microorganisms present, the state of the uterus, and its innate defense mechanism all affect the clinical indications of uterine infection [11]. Immunological monitoring is frequently carried out to identify pathogenic, diagnostic, and prognostic signs in inflammatory disorders; this process is noteworthy for the discovery of novel therapeutic or diagnostic targets [12]. The most effective way to treat postpartum metritis in livestock, which includes endometritis, postpuerperial metritis, toxic puerperial metritis, and pyometra, is still up for debate [13]. Clinical investigations that assessed non-antibiotic therapeutic agents, dosages, routes of administration, and timing in relation to days after parturition had inconsistent results. Due to the rise in disease resistance, antibiotic residues and preventative drugs are no longer preferred [14]. Utilizing host genetic resistance, which is unrestricted in the broadest sense by these disadvantages, is the solution for the low-cost and efficient control of various diseases [15]. Unfortunately, the identification and characterization of numerous gene loci is required to properly treat these disorders [16].

DNA testing for the major genes influencing fertility and inheritance patterns has revealed scientifically that these genes have a significant potential to enhance reproductive success in sheep worldwide [17]. In addition to being a useful tool for assessing the physiological status of individual animals, diagnosing metabolic disorders, determining clinical nutritional balances, determining deficit conditions, monitoring treatment, and making prognoses, measuring gene expression is a reliable method for assessing the health status of sheep [18]. Remarkably, significant numbers of essential genes are found in sheep populations [19,20]. These important genes have been linked to certain characteristics of the general population related to reproduction, illness, or productivity [21]. Additionally, investigations on gene expression showed that thousands of genes involved in immune function were exhibited by animals with high resistance [22,23].

The immunological alterations and genetic variants associated with postpartum endometritis in sheep are equally little understood, and few reliable diagnostic and prognostic instruments are available to help us in the development of our preventive and treatment strategies [24,25]. It is imperative to underscore that further investigation is required to detect any pertinent immunological markers linked to reproductive concerns, which may ascertain whether sheep exhibit resistance to them. By real-time PCR and PCR-DNA sequencing practices, the aim of this study was to evaluate the potential immunological, metabolic, and antioxidant genes for their efficacy in endometritis prevalence expectation and following in Ossimi sheep.

## 2. Material and Methods

### 2.1. Investigated Ewes and Research Samples

One hundred Ossimi sheep, possessing a 3.5-year age average and a body weight of $46 \pm 2.4$ kg, were used in the current study. Animals were sourced from commercial farms in Egypt's northern Nile Delta. The used ewes were immunized and were from the same group and litter size. Based on their postpartum health status, the examined sheep were separated into two groups of identical size (50 ewes each). The ewes under investigation had their body temperatures, pulse rates, respiration rates, and uteri examined in the period of 30 days postpartum.

The control group (CG) was made up of clinically healthy ewes who had normal lambing and postpartum stages (i.e., normal feed intake, body temperature, and no uterine discharge). Ewes with clinical endometritis (pyrexia, persistent colored uterine discharge with foul odor, anorexia, depression) made up the second group. Both groups were chosen based on breeding history and clinical signs observed by knowledgeable veterinarians. Each animal was kept in a separate stall with free access to water. Blood samples were aseptically taken from each ewe's jugular vein. The blood samples were taken before the immediate medical treatment for endometritis affected ewes. The freshly collected blood samples were put in tubes containing the anticoagulant EDTA to extract DNA and RNA. At the site of collection, an ice box was taken for preservation of blood samples; then, the collected samples were urgently sent to laboratory to avoid DNA and RNA degradation. Once the samples arrived at the laboratory, they were immediately extracted for DNA and RNA or kept at −80 °C for further examination. The Princess Nourah bint Abdulrahman University Animal Care and Use Committee (MU-ACUC), with agreement number PNURS.R.23.10.30, gave its clearance before the animal experiment was carried out.

### 2.2. DNA Extraction and Amplification

Nucleic acid from the genome was separated using whole blood and the genetic material JET entire blood genomic nucleic acid isolation kit in accordance with the manufacturer's instructions (Thermo Scientific, Vilnius, Lithuania). Nanodrop was utilized to examine DNA that possessed high purity and intensity. Immune (*A2M*, *TLR2*, *TGF-β*, *IRAK3*, *HMGB1*, *FCAMR*, and *iNOS*), metabolic (*ADAMTS20*, *KCNT2*, *MAP3K4*, *FKBP5*, and *RXFP1*) and antioxidant (*SOD*, *CAT*, *NDUFS5*, and *HMOX1*) genes were cloned in vitro. The PubMed *Ovis aries* genome was used to generate the amplification-related oligonucleotide sequences. A listing of primers that were utilized in the PCR can be found in Table 1.

**Table 1.** Oligonucleotide primers for the immune, metabolic, and antioxidant genes used to study genetic polymorphisms.

| Gene of Investigation | Sense | Antisense | Annealing Temperature (°C) | Size of PCR Product (bp) |
|---|---|---|---|---|
| *A2M* | 5′-TCAAGAGGTAATGTTCCTCAC-3′ | 5′-CACCTCTTCTTCCAAGATGGTG-3′ | 58 | 448 |
| *TLR2* | 5′-AATCTTGAATATTTGGACTTA-3′ | 5′-CAGGCTGCCTCAGATGCAGAAT-3′ | 58 | 340 |
| *TGF-β* | 5′-CTGACGCCTGGCCGGCCGGTCG-3′ | 5′-GAGAGAGCAACACAGGTTCAG-3′ | 60 | 394 |
| *IRAK3* | 5′-CCGCGGTTGTGTAACGGCTCCG-3′ | 5′-GATGACTCTGCTCTATAGGACT-3′ | 58 | 431 |
| *HMGB1* | 5′-CAGTGGAAGCCCAGAGTTTTA-3′ | 5′-ACTCTGAGAAGTTGACTGAAG-3′ | 60 | 430 |
| *FCAMR* | 5′-AGCGCGAAGCGACCTGATGG-3′ | 5′-TACCGGCACCGGATGGTCACA-3′ | 60 | 386 |
| *iNOS* | 5′-CAGGAACCTACCAGCTGACG-3′ | 5′-TCCAGCCCAGGTCGATGCACA-3′ | 58 | 360 |
| *ADAMTS20* | 5′-AGGTCGGCTTGCTTCCTCGCG-3′ | 5′-TCACCACTTCGTAGGAGGCCAG-3′ | 60 | 465 |
| *KCNT2* | 5′-CGGCCTCAGCTTCCGGCTGG-3′ | 5′-CTAGTAGCACTCGGATTATGT-3′ | 58 | 442 |
| *MAP3K4* | 5′-TCTGATCCTGAGGACTATTCA-3′ | 5′-TCGGCTGCAGCAATGCTAAGCT-3′ | 60 | 398 |
| *FKBP5* | 5′-ATCAAGGCATGGGACATTGG-3′ | 5′-ATACATTGTTCCTCACGCTGCA-3′ | 58 | 389 |
| *RXFP1* | 5′-TAGCATACAACCACCGTGAGA-3′ | 5′-CTCATCACACTCGAGCTCTA-3′ | 58 | 435 |
| *SOD* | 5′-CTGCCTGAGGAGCTGCACAC-3′ | 5′-CGTTGTGAATCGTCCATCCAAC-3′ | 58 | 385 |
| *CAT* | 5′-CTGCAGCGCCGCTCAGACAC-3′ | 5′-CTGTGTCAGCTGAGCCTGATTC-3′ | 60 | 397 |
| *NDUFS5* | 5′-CTGAGGATCGTCTAGGAGAG-3′ | 5′-TGTCTCTTGATGGCATTCAGAC-3′ | 60 | 415 |
| *HMOX1* | 5′-CTACGCAGCCGCAGGATGGAGC-3′ | 5′-ACCTCCTGGAGTCGCTGAACA-3′ | 58 | 377 |

*A2M* = Alpha-2-macroglobulin; *TLR2* = Toll-like receptor 2; *TGF-β* = Transforming growth factor beta; *IRAK3* = Interleukin 1 receptor-associated kinase 3; *HMGB1* = High-mobility group box 1; *FCAMR* = Fc alpha and Mu receptor; *iNOS* = Inducible nitric oxide synthase; *ADAMTS20* = ADAM metallopeptidase with thrombospondin type 1 motif 20; *KCNT2* = Potassium sodium-activated channel subfamily T member 2; *MAP3K4* = Mitogen-activated protein kinase kinase 4; *FKBP5* = FKBP prolyl isomerase 5; *RXFP1* = Relaxin family peptide receptor 1; *SOD* = Superoxide dismutase; *CAT* = Catalase; *NDUFS5* = NADH:ubiquinone oxidoreductase subunit s5; and *HMOX1* = Heme oxygenase-1.

A thermal cycler with an ultimate capacity of 50 microliters was used to process the polymerase chain cloning mixture. The following components were used in each reaction: 20 microliters of d.d. water, 3 microliters of genetic material with a 100 ng concentration, 1 microliter of every primer (20 pmol), and 25 microliters of the master mixture (Jena Bioscience, Jena, Germany). An initial temperature of 95 °C for the separation of the DNA helix was achieved for four minutes. The reaction continued for 35 cycles; it included one-minute rounds of denaturation at 95 °C, annealing cycles lasting one minute each based on the temperature variety presented in Table 1, 45 s to 30 s rounds of elongation at 72 °C followed by seven additional minutes of last extension at 72 °C. The PCR products were maintained at 4 °C. PCR segment configurations were analyzed under UV light as part of a gel certification method, and a 2% agarose (Bio-Rad, Hercules, CA, USA) gel electrophoresis was utilized to yield visible outcomes.

### 2.3. Identifying Polymorphism

Jena Bioscience #pp201s/Munich of Hamburg, Germany provided purification processes prior to DNA sequencing to dispose of primer dimmers, nonspecific bands, and other impurities to yield the suitable amplification for the expected scope [26]. Since the Nanodrop Q5000 UV-Vis spectrophotometer supplied adequate quality and good concentrations, it was used to assess PCR output [27]. To find SNPs in healthy and endometrtis-affected ewes, sequence analysis was performed on the PCR data from forward and reverse primer amplification. On an ABI 3730XL DNA sequencer (Applied Biosystems, Waltham, MA, USA), the enzyme chain terminator method outlined by Sanger et al. [28] was used to sequence PCR data. An evaluation of DNA analysis results was conducted using BLAST 2.0 and Chromas 1.45 [29]. When the immunological, metabolic, and antioxidant indicator PCR findings were compared to the reference gene sequences made available by GenBank, polymorphisms were discovered. The MEGA6 program was used to detect variations in the groups of amino acids amongst the studied genes based on sequence matching among the investigated sheep [30].

### 2.4. Levels of Immune, Metabolic, and Antioxidant Gene Transcripts

According to the manufacturer's recommendations, the RNA extracted from blood samples obtained from the sheep under examination was entirely extracted using the Trizol solution (RNeasy Mini Ki, 74104, Product No., Venlo, The Netherlands). The amount of RNA that was separated was quantified and confirmed by means of a NanoDrop® ND-1000 spectrophotometer. Complementary nucleic acid for each sample was prepared according to the producer's procedure (Waltham, MA, USA: Thermo Fisher, Product No. EP0441). To evaluate the immunological, metabolic, and antioxidant gene expression patterns, quantitative PCR and SYBR Green PCR Master Mix were utilized (2× SensiFastTM SYBR, Bio-line, CAT No. Bio-98002). The SYBR Green PCR Master Mix (Quant-titect SYBR green PCR reagent, Toronto, ON, Canada; Catalogue No. 204141) was applied to determine the proportion of mRNA that is present in every sample.

Table 2 shows that the *Ovis aries* genome from PubMed was employed to produce the primer oligonucleotides for sense and antisense. The *β. actin* gene supplied a constitutive normalization reference. In total, 3 microliters of RNA, 4 microliters of 5xTrans Amp buffer, 0.25 microliters of reverse transcriptase, 0.5 microliters of each primer, 12.5 microliters of 2× Quantitect SYBR green PCR master mix, and 8.25 microliters of RNase-free water made up the 25-microliter reaction mixture, which contained only RNA. Following that, the reaction mixture was heated to 55 °C for 30 min, followed by 40 cycles of 15 s at 95 °C for preliminary denaturation, primer hybridizing temperatures as shown in Table 2, and a 1 min extension at 72 °C. All of these procedures were carried out inside the PCR cycler. Once the PCR product was amplified, a melting curve analysis was carried out to show its specificity. The differences in each gene's expression were investigated using the $2^{-\Delta\Delta Ct}$ method, which compares each marker's mRNA level in the test sample to that of the *β. actin* gene [31];

ΔCt for sample = Ct for sample—Ct for the reference (*β. actin*) gene;
ΔΔCt = ΔCt for sample—ΔCt for control.

**Table 2.** Real-time PCR primers made of oligonucleotides that are sense and antisense for immune, metabolic, and antioxidant genes under study.

| Investigated Marker | Primer | Product Size (bp) | Annealing Temperature (°C) | GenBank Isolate | Origin |
|---|---|---|---|---|---|
| *A2M* | F5′-ACCCAAGATATGTCCACAGCC-3 R5′-CAGCTAAGGCTGGAGAAGCTA-3′ | 80 | 58 | XM_012175446.4 | |
| *TLR2* | F5′-TGACAAGAAGGCCGTTCCCC-3 R5′-CTCCAGGTAGGTCCTGGTGTT-3′ | 70 | 58 | DQ890157.1 | |
| *TGF-β* | F5′-GCCCTGTCCCTACATCTGGA-3′ R5′-GTAGTACACGATGGGCAGGG-3′ | 136 | 60 | NM_001009400.2 | |
| *IRAK3* | F5′-ACTGTGCTCTGTCGTCTGTG-3′ R5′-TGCTGGTCATGCTTATGGCA-3′ | 143 | 58 | XM_027967477.2 | |
| *HMGB1* | F5′-GGTTTCTTGATCCATTTCCCTGC-3′ R5′-AGATCACGGTCCCCGAAAAC-3′ | 215 | 60 | XM_042254827.1 | |
| *FCAMR* | F5′-CAGCATGAACCTGACGGTCT-3′ R5′-CAGCCGTATCCCATCCTGTC-3′ | 176 | 60 | XM_042257020.1 | |
| *iNOS* | F5′-AAGTGGTATGCTCTGCCAGC -3′ R5′-CCCATGTACCACCCGTTGAA-3′ | 92 | 58 | AF223942.1 | |
| *ADAMTS20* | F5′-TGCTGACAGAGAATGCCAGG -3′ R5′-GCATGAAGCTCACAGGGTCT-3′ | 225 | 56 | XM_004006435.5 | |
| *KCNT2* | F5′-AGGACGCAAAAGCCTATGGA -3′ R5′-AATGCATGTCTGGCGGGTTA-3′ | 165 | 60 | XM_027976255.2 | Present Research |
| *MAP3K4* | F5′-AGCCATTGAGCCTTCGTTCA-3′ R5′-GCTGTTCCGATGAATGGCTG-3′ | 133 | 58 | XM_042253698.1 | |
| *FKBP5* | F5′-AGATAGAGGAGTTTTAAAGGCCAA -3′ R5′-GCATTCGAGGGAATTTTGGGG-3′ | 146 | 58 | XM_042237057.1 | |
| *RXFP1* | F5′-GGACAACTGCGGAGACATCA-3′ R5′-AGACCCGACCAAGCATTCAG-3′ | 124 | 56 | XM_012097572.3 | |
| *SOD* | F5′-AGAATGCAGAGTGGGAGGAAC-3′ R5′-GCTGTATTCCAGAAACTGGGC-3′ | 188 | 60 | XR_006060140.1 | |
| *CAT* | F5′-ATTGCGGGCCATCTGAAAGA-3′ R5′-GCACATAGGTGTGAACTGCG-3′ | 154 | 60 | XM_004016396.5 | |
| *NDUFS5* | F5′-AGGAGAAAAGGACACAGCGG-3′ R5′-GCATGGCAGTCTCCCAATTTC-3′ | 144 | 58 | XM_027968234.2 | |
| *HMOX1* | F5′-CAAGCGCTATGTTCAGCGAC-3′ R5′-GCTTGAACTTGGTGGCACTG-3′ | 206 | 56 | XM_045162483.1 | |
| *β. actin* | F5′-AATTCCATCATGAAGTGTGAC-3′ R5′-GATCTTGATCTTCATCGTGCT-3′ | 150 | 58 | KU365062.1 | |

*A2M* = Alpha-2-macroglobulin; *TLR2* = Toll-like receptor 2; *TGF-β* = Transforming growth factor beta; *IRAK3* = Interleukin 1 receptor-associated kinase 3; *HMGB1* = High-mobility group box 1; *FCAMR* = Fc alpha and Mu receptor; *iNOS* = Inducible nitric oxide synthase; *ADAMTS20* = ADAM metallopeptidase with thrombospondin type 1 motif 20; *KCNT2*= Potassium sodium-activated channel subfamily T member 2; *MAP3K4* = Mitogen-activated protein kinase kinase 4; *FKBP5* = FKBP prolyl isomerase 5; *RXFP1* = Relaxin family peptide receptor 1; *SOD* = Superoxide dismutase; *CAT* = Catalase; *NDUFS5* = NADH:ubiquinone oxidoreductase subunit s5; and *HMOX1* = Heme oxygenase-1.

### 2.5. Statistical Analysis

**H$_0$:** *Polymorphisms in immune, metabolic, and antioxidant genes are not associated with incidence of postparturient endometritis in Ossimi sheep.*

**H$_A$:** *Polymorphisms in immune, metabolic, and antioxidant genes are associated with incidence of postparturient endometritis in Ossimi sheep.*

Using Fisher's exact test analysis, the considerable distribution of SNPs for the indicated genes could be seen in the examined ewes ($p < 0.01$). Statistical Package for Social Science (SPSS) version 17 and the *t*-test (SPSS Inc., Chicago, IL, USA) were applied to determine the statistical significance of the difference between healthy and endometritis-affected ewes. The data were presented using mean and standard error (Mean ± SE). At $p < 0.05$, disparities were evidently substantial.

## 3. Results

### 3.1. Genetic Polymorphisms of Immune, Metabolic, and Antioxidant Genes

SNP changes in amplified DNA nucleotides linked to endometiris were discovered in the PCR-DNA sequencing data for *A2M* (465 bp), *TLR2* (465 bp), *TGF-β* (465 bp), *IRAK3* (465 bp), *HMGB1* (465 bp), *FCAMR* (465 bp), *iNOS* (465 bp), *ADAMTS20* 465 bp, *KCNT2* (465 bp), *MAP3K4* (465 bp), *FKBP5* (465 bp), *RXFP1* (465 bp), *SOD* (465 bp), *CAT* (465 bp), *NDUFS5* (465 bp), and *HMOX1* (465 bp). The DNA sequence differences between the GenBank-obtained reference gene sequences and the immune, metabolic, and antioxidant indicators evaluated in the ewes under study were utilized to confirm each SNP that was found (Figures S1–S16).

In both normal and endometritis-affected ewes, Table 3 shows the scattering of an inherited modification type and a single base variation for markers related to immunity, metabolism, and antioxidants. The examination of SNP Fisher's exact test showed that the ewes with endometritis and those without it had significantly different frequencies of the markers under investigation ($p < 0.01$). Exonic area mutations were seen in all immunological, metabolic, and antioxidant markers under study, resulting in distinct coding DNA sequences in sheep affected by endometritis compared to normal ones. The sequencing of immune, metabolic, and antioxidant genes yielded 37 SNPs, of which 11 were synonymous and 26 non-synonymous.

**Table 3.** Dispersion of immunological, metabolic, and antioxidant indicators in endometritis and healthy ewes with a single base differential and a potential genetic amendment.

| Gene | SNPs | Healthy $n = 50$ | Endometritis $n = 50$ | Total $n = 100$ | Kind of Genetic Change | Amino Acid Order and Nature |
|---|---|---|---|---|---|---|
| *A2M* | G94A | -/50 | 31/50 | 31/50 | Non-synonymous | 32 G to S |
| *TLR2* | G128A | 28/50 | -/50 | 28/50 | Non-synonymous | 43 G to E |
| *TGF-β* | G73C | 19/50 | -/50 | 19/50 | Non-synonymous | 25 G to R |
| | T93C | 33/50 | -/50 | 33/50 | Synonymous | 31 G |
| | C354T | -/50 | 24/50 | 24/50 | Synonymous | 118 S |
| *IRAK3* | G48T | -/50 | 35/50 | 35/50 | Non-synonymous | 16 R to S |
| | A148C | -/50 | 17/50 | 17/50 | Non-synonymous | 50 S to R |
| | C352T | -/50 | 19/50 | 19/50 | Non-synonymous | 118 P to S |
| | A404G | 41/50 | -/50 | 41/50 | Non-synonymous | 135 N to S |
| *HMGB1* | G196C | 32/50 | -/50 | 32/50 | Non-synonymous | 66 V to L |
| *FCAMR* | A82G | 17/50 | -/50 | 17/50 | Non-synonymous | 28 S to G |
| *iNOS* | C77T | 39/50 | -/50 | 39/50 | Non-synonymous | 26 S to L |
| | T158C | 24/50 | -/50 | 24/50 | Non-synonymous | 53 V to A |
| *ADAMTS20* | A37G | -/50 | 32/50 | 32/50 | Non-synonymous | 13 I to V |
| | C383G | 27/50 | -/50 | 27/50 | Non-synonymous | 128 S to C |
| *KCNT2* | A46G | 35/50 | -/50 | 35/50 | Non-synonymous | 16 T to A |
| | A169T | -/50 | 41/50 | 41/50 | Non-synonymous | 57 T to S |
| | A399G | 29/50 | -/50 | 29/50 | Synonymous | 133 L |
| *MAP3K4* | T111C | -/50 | 33/50 | 33/50 | Synonymous | 37 N |
| | G208A | -/50 | 23/50 | 23/50 | Non-synonymous | 70 V to M |
| *FKBP5* | T281C | 37/50 | -/50 | 37/50 | Non-synonymous | 94 M to T |
| *RXFP1* | C109A | 24/50 | -/50 | 24/50 | Non-synonymous | 37 L to I |
| | T126C | 19/50 | -/50 | 19/50 | Synonymous | 42 F |

**Table 3.** *Cont.*

| Gene | SNPs | Healthy $n = 50$ | Endometritis $n = 50$ | Total $n = 100$ | Kind of Genetic Change | Amino Acid Order and Nature |
|---|---|---|---|---|---|---|
| *RXFP1* | G235A | -/50 | 27/50 | 27/50 | Non-synonymous | 79 V to M |
| | C285T | -/50 | 18/50 | 18/50 | Synonymous | 95 N |
| | C353T | 38/50 | -/50 | 38/50 | Non-synonymous | 118 T to I |
| *SOD* | G46C | 23/50 | -/50 | 23/50 | Non-synonymous | 16 A to P |
| | G97A | 17/50 | -/50 | 17/50 | Non-synonymous | 33 G to R |
| | A320G | 43/50 | -/50 | 43/50 | Non-synonymous | 107 N to S |
| | G350A | 30/50 | -/50 | 30/50 | Non-synonymous | 117 R to K |
| *CAT* | T63C | 29/50 | -/50 | 29/50 | Synonymous | 31 P |
| | C246T | -/50 | 33/50 | 33/50 | Synonymous | 82 H |
| *NDUFS5* | G50T | -/50 | 26/50 | 26/50 | Non-synonymous | 17 G to V |
| | G168A | -/50 | 34/50 | 34/50 | Synonymous | 56 L |
| *HMOX1* | C108T | 23/50 | -/50 | 23/50 | Synonymous | 36 A |
| | T237C | -/50 | 19/50 | 19/50 | Synonymous | 79 Y |
| | C352T | 37/50 | -/50 | 37/50 | Non-synonymous | 118 R to C |

Single-base variance scattering for immune, metabolic, and antioxidant markers in normal and endometeritis ewes disclosed an extremely significant discrepancy ($p < 0.01$) according to Fisher's exact analysis. *A2M* = Alpha-2-macroglobulin; *TLR2* = Toll-like receptor 2; *TGF-β* = Transforming growth factor beta; *IRAK3* = Interleukin 1 receptor-associated kinase 3; *HMGB1* = High-mobility group box 1; *FCAMR* = Fc alpha and Mu receptor; iNOS = Inducible nitric oxide synthase; *ADAMTS20* = ADAM metallopeptidase with thrombospondin type 1 motif 20; *KCNT2* = Potassium sodium-activated channel subfamily T member 2; *MAP3K4* = Mitogen-activated protein kinase kinase 4; *FKBP5* = FKBP prolyl isomerase 5; *RXFP1* = Relaxin family peptide receptor 1; *SOD* = Superoxide dismutase; *CAT* = Catalase; *NDUFS5* = NADH: ubiquinone oxidoreductase subunit s5; and *HMOX1* = Heme oxygenase-1. A = Alanine; C = Cisteine; E = Glutamic acid; F = Phenylalanine; G = Glycine; H = Histidine; I = Isoleucine; K = lysine; L = Leucine; M = Methionine; N = Asparagine; P = Proline; R = Argnine; S = Serine; T = Threonine; V = Valine; and Y = Tyrosine.

For the immune indicators, the *A2M* gene (448 bp), substitutional alteration G94A SNP led to the occurrence of G32S. When 340 bp for the *TLR2* gene was sequenced, one recurrent SNP was discovered, where the G128A SNP caused the non-synonymous mutation G43E. DNA sequencing was applied to discover three recurring SNPs in the 394 bp of the *TGF-β* gene; two of them, T93C and C354T, led to synonymous mutations, 31G and 118S, respectively. By contrast, the amino acid G25R was substituted as a result of the non-synonymous mutation brought forth by the G73C SNP. The *IRAK3* gene (431 bp) enclosed four non-synonymous SNPs, G48T, A148C, C352T, and A404G, which produced amino acids R16S, S50R, P118S, and N135S to be exchanged, correspondingly. DNA sequencing was used to identify a single recurrent SNP (430 bp) in the *HMGB1* gene; G196C produced the non-synonymous mutation V66L. DNA sequencing was employed to recognize a single repeated SNP (386 bp) in the *FCAMR* gene; A82G produced non-synonymous mutation S28G. Two recurring non-synonymous SNPs that were found to be present in the *iNOS* gene (399 bp) were C77T and T158C. These SNPs produced S26L and V53A amino acids, respectively.

With regard to metabolic indicators such as the *ADAMTS20* gene (465 bp), two non-synonymous mutations, I13V and S128C, occurred as a result of A37G and C383G SNPs, respectively. The *KCN2* gene (442 bp) contains three repeating SNPs that were identified by DNA sequencing; two of them, A46G and A169T, produced non-synonymous mutations, T16A and T57S, respectively. On the other hand, amino acid 133L was produced by the synonymous mutation caused by SNP A399G. Two repetitive SNPs were found when the *MAP3K4* (398 bp) gene was sequenced. T111C involved the synonymous mutation 37N, and G208A involved the non-synonymous mutation that changed the amino acid V70M. Upon sequencing the 389 bp of the *FKBP5* gene, a recurrent SNP was discovered,

T281C SNP, which resulted in the non-synonymous mutation M94T. Five regular SNPs in the *RFBP1* gene (435 bp) were found by DNA sequence; three of these, C109A, G235A, and C353T, resulted in non-synonymous changes, L37I, V79M, and T118T, respectively. However, synonymous mutations brought about by T126C and C285T SNPs resulted in the production of amino acids 42F and 95N, respectively.

In terms of antioxidant markers, four recurring non-synonymous SNPs were identified in the 385 bp nucleotide categorization of the *SOD* gene: A16P, G33R, N107S, and R117K. These altered amino acids were caused by the G46C, G97A, A330G, and G350A SNPs, respectively. Two frequently occurring synonymous SNPs, T63C and C246T, were found in the 397 bp DNA structure of the *CAT* marker and were linked to amino acids 31P and 82H, respectively. When the 415 bp DNA sequence of the *NDUFS5* gene was analyzed, two recurrent SNPs were discovered: G50T carried the non-synonymous mutation G17V and G168A included the synonymous mutation 56L. The *HMOX1* gene (377-bp) has three repeated SNPs that were established by DNA sequencing; one of these, T273C, led to synonymous mutation, 79Y, while amino acids T36A and R118C, respectively, were substituted as a result of a non-synonymous mutation caused by SNPs C108T, and C352T.

### 3.2. Tendencies of Immune, Metabolic, and Antioxidant Marker mRNA Levels

The transcript levels related to immunology, metabolism, and antioxidants in healthy and endometritis ewes are displayed in Figure 1, Figure 2 and Figure 3, respectively. The expression levels of *A2M*, *TLR2*, *IRAK3*, *HMGB1*, *FCAMR*, *iNOS*, *ADAMTS20*, *KCNT2*, *MAP3K4*, *FKBP5*, *RXFP1*, and *HMOX1* were significantly up-regulated in endometritis ewes. On the other hand, genes encoding *TGF-β*, *SOD*, *CAT*, and *NDUFS5* were down-regulated. *TLR2* had the maximum potential quantity of mRNA for endometritis ewes ($2.85 \pm 0.13$), whereas *TGF-β* had the lowest amount ($0.53 \pm 0.08$) of each gene. *SOD* had the highest potential level of mRNA ($2.18 \pm 0.13$) of all the genes analyzed in the healthy ewes, while *FCAMR* had the lowest amount ($0.36 \pm 0.14$).

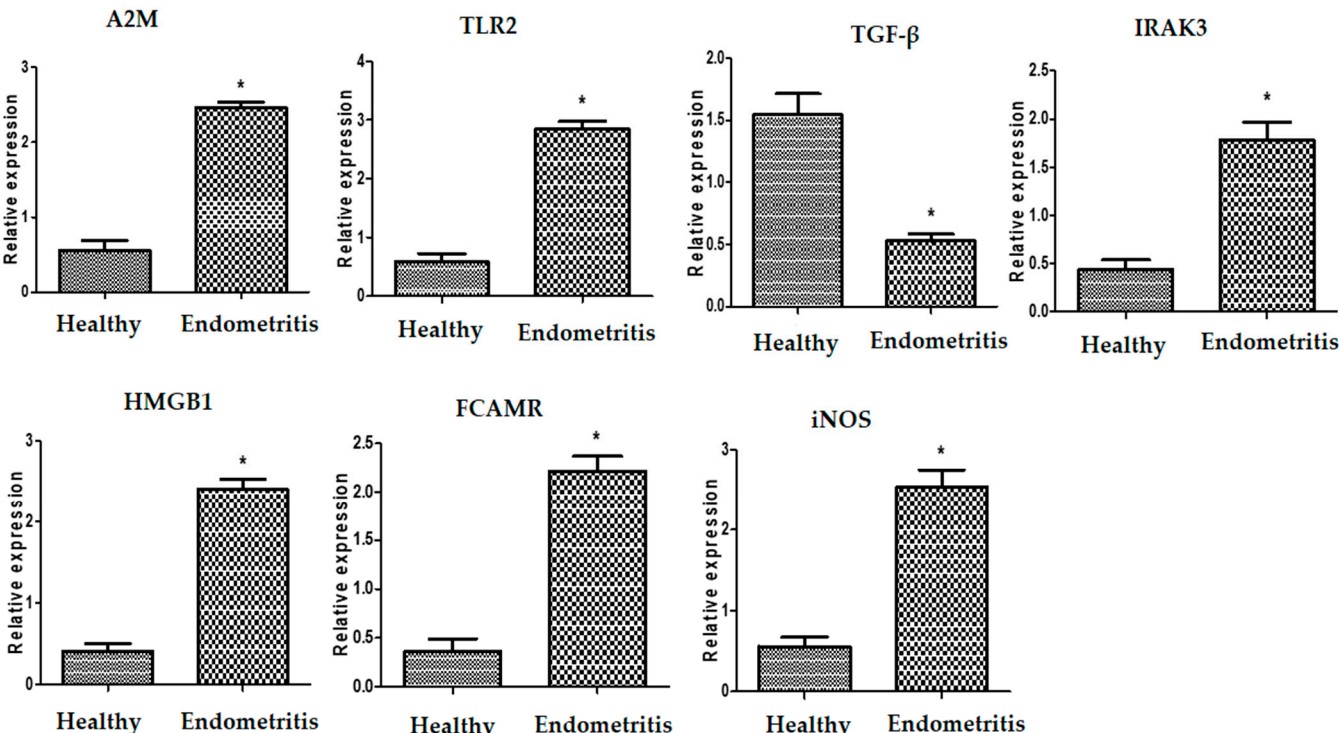

**Figure 1.** Different transcript intensities of immune genes between normal and endometritis ewes. When $p < 0.05$, the asterisk (*) indicates significance.

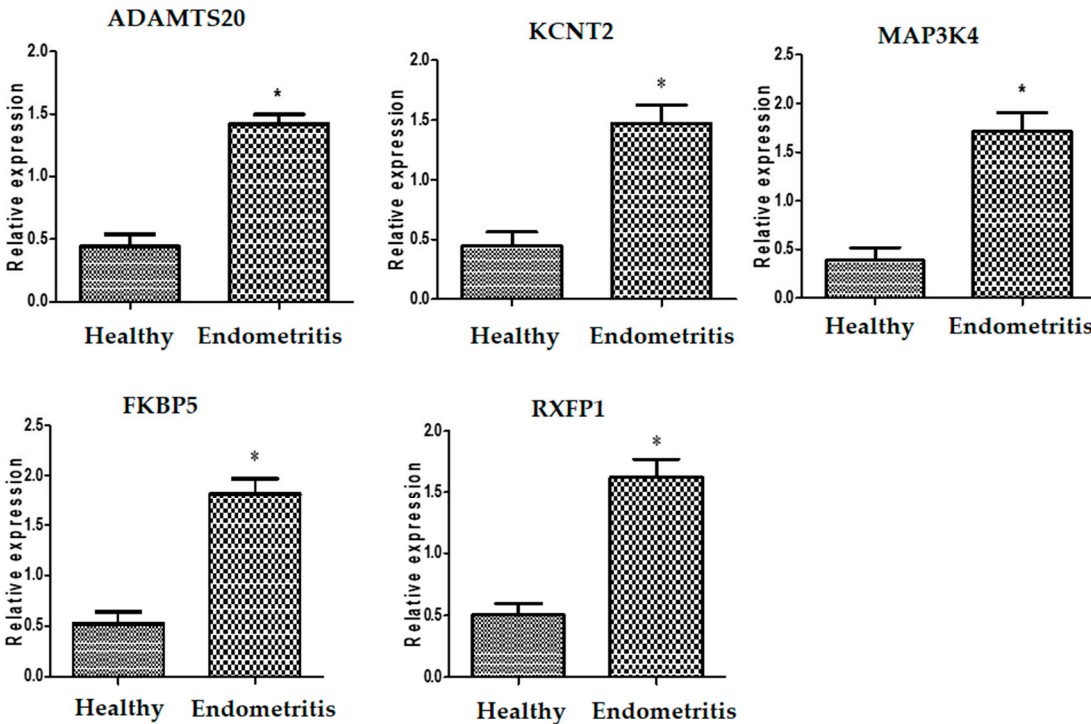

**Figure 2.** Different transcript intensities of metabolic genes between normal and endometritis ewes. When $p < 0.05$, the asterisk (*) indicates significance.

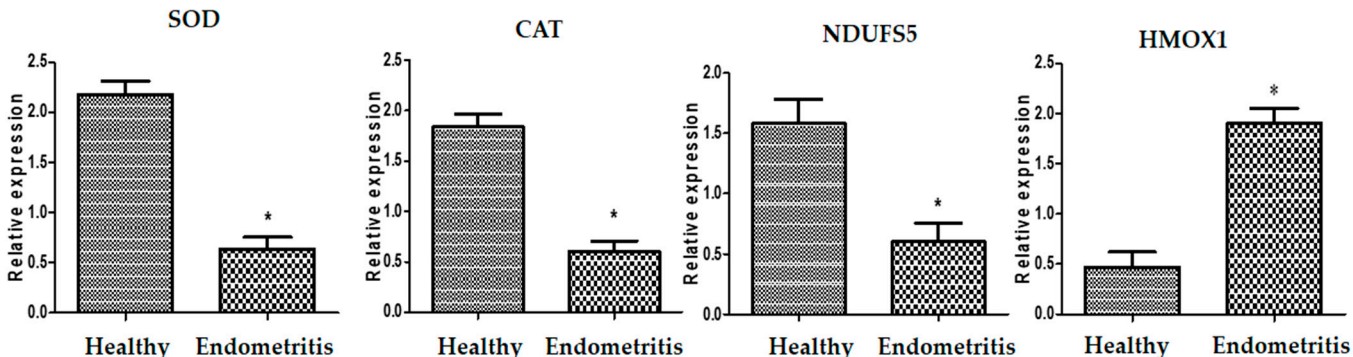

**Figure 3.** Different transcript intensities of antioxidant genes between normal and endometritis ewes. When $p < 0.05$, the asterisk (*) indicates significance.

## 4. Discussion

### 4.1. Variations in Immune, Metabolic, and Antioxidant Genes Associated with Susceptibility to Endometritis

In order to fully eradicate diseased animals or utilize disease-resistant livestock, it is imperative to comprehend the underlying mutations and interactions that lead to genetic resistance [32]. Sequencing of the amplified PCR products revealed variations in the immunological, metabolic and antioxidant genes in both endometritis and healthy Ossimi ewes. The findings show that there are differences between the SNPs for the two categories. It is crucial to stress that the differences found and the materials easily accessible in this context provide new insights into the markers under investigation when matched to the relatable sequence acquired from GenBank.

Employing genome-wide association analysis for our investigated markers, recent studies have identified distinct genes that are specifically linked to the prevalence of bovine endometritis [33]. Nevertheless, no investigation has yet examined the connection between endometritis risk and the SNPs in these genes. We are the first to demonstrate this

connection with gene sequences from published PubMed *Ovis aries*. As far as we are aware, no previous research has examined the relationship between postparturient endometritis in Ossimi sheep and the variation in immunological (*A2M*, *TLR2*, *TGF-β*, *IRAK3*, *HMGB1*, *FCAMR*, and *iNOS*), metabolic (*ADAMTS20*, *KCNT2*, *MAP3K4*, *FKBP5*, and *RXFP1*) and antioxidant (*SOD*, *CAT*, *NDUFS5*, and *HMOX1*) markers.

The candidate gene mode could be used to evaluate the authenticity of sheep postpartum endometritis. In Barki sheep, the incidence of postpartum sickness was also linked to *TLR4* and *SOD* SNPs [34]. Genetic polymorphisms of immunological, metabolic, and antioxidant indicators for postparturient endometritis were investigated in Holstein dairy cattle in order to increase an enhanced considerate of resistance and sensitivity of inflammatory reproductive disorders in cattle [35]. The nucleotide sequences of cows with endometritis and those in good health differed. Sequence analysis connected polymorphisms in the *TLR2* and *TLR4* indicators in river buffalo with endometritis [36,37]. Furthermore, in Italian buffaloes, immunological biomarker mRNA levels and DNA polymorphisms were demonstrated to be potential indicators of postpartum endometritis susceptibility [38]. DNA polymorphisms and the immunological and antioxidant gene expression profile in goats were shown to be indicators for endometritis susceptibility or tolerance in Baladi goats [39].

### 4.2. Gene Expression Trend of Immune, Metabolic, and Antioxidant Markers

Our hypothesis was that genetic diversity in sheep's transcriptional response to the condition's incidence could influence the development of postpartum endometritis. Real-time PCR was carried out to evaluate the levels of the immune, metabolic, and antioxidant genes in both normal and endometritis-affected ewes. According to our research, endometritis ewes had considerably higher expression levels of *A2M*, *TLR2*, *IRAK3*, *HMGB1*, *FCAMR*, *iNOS*, *ADAMTS20*, *KCNT2*, *MAP3K4*, *FKBP5*, *RXFP1*, and *HMOX1*. Conversely, there was a down-regulation of the genes that encode *TGF-β*, *SOD*, *CAT*, and *NDUFS5*. The mRNA levels of these markers and their correlation with the incidence of postpartum endometritis in sheep were examined for the first time in this investigation. We performed SNP genetic markers and gene expression to analyze genetic variation, overcoming the limitations of previous investigations. As a result, the tools regulating the examined genes were well assumed by both normal and endometritis ewes.

In reference to the sheep's gene expression profile for immunological and antioxidant markers, postpartum sheep exhibited significantly greater levels of mRNA for *IL1-ß*, *TNF alpha*, *IL5*, *IL6*, *TLR4*, and *Tollip*, as reported by Darwish et al. [34]. *SOD* and *CAT* gene levels, however, were considerably lower.

Livestock was used to appraise the gene manifestation method of immunological, metabolic, and antioxidant indicators. Compared to cows without endometritis, inflamed dairy cows showed higher transcript intensities of immunological genes (*CCL5*, *CXCL8*, *IL6*, and *IL1B*) in their bovine endometrium [40]. Endometritis-affected cows expressed significantly more *NCF4*, *LITAF*, *TLR4*, *OXSR1*, *TLR7*, *TNF-α*, *TKT*, *RPIA*, and *AMPD1* genes than resistant cows did. In the meantime, there was a significant drop in the expression of the *GST*, *ATOX1*, and *IL10* genes [35]. Dairy calves with subclinical endometritis had specific transcript levels of *C2*, *LTF*, *TRAPPC13*, *C3*, and *PF4* in their blood and endometrium [41].

According to a study on cytokine gene expression in the uterus of *Bubalus bubalis* linked to endometritis, buffaloes with endometritis had greater levels of *IL-1* and *IL-6* gene expression than control animals. On the other hand, the expressions of TNF-α and *IL-10* showed decreased mRNA by 0.4- and 0.2-fold, respectively [42]. While *GPX4*, *GST*, *SOD3*, *CAT*, and *ATOX1* gene patterns induced an opposing trend, the levels of *PRLR*, *LTF*, *CLA-DRB3.2*, *beta defensin*, *TLR2*, *TLR4*, and *CCL5* markers were considerably up-surged in individuals impacted with postparturient endometritis goat compared to those of tolerant ones [39].

### 4.3. The Efficacy of the Genes Examined as Possible Candidates for Endometritis Susceptibility

4.3.1. Role of Examined Immune Markers

Apart from C3, C4, and C5, A2M belongs to the alpha-macroglobulin protein family [43]. Furthermore, it stimulates T cell and macrophage proliferation [44]. Upon activation, TLR regulates the production of several chemokines and pro-inflammatory cytokines, thereby inducing innate and acquired immune responses and contributing to the enhancement of neutrophil recruitment [45]. TLR2 is one of the key form recognition receptors and is required to initiate the inflammatory and immune response [46].

TGF-β is a multifunctional peptide that belongs to a family of cytokines that regulate proliferation, differentiation, adhesion, migration, and immunological modulation. IRAK-M, also known as IRAK3, is mostly expressed in monocytes and macrophages [47]. IRAK3 is essential for controlling the innate immune system's TLR signaling pathways [48]. In Han Chinese population, *IRAK3* single-nucleotide polymorphisms were found to positively correlate with susceptibility to sepsis [49]. In clinical specimens from septic patients, *IRAK3* transcript expression is noticeably elevated.

A widely conserved protein with important biological roles is HMGB1. It can stimulate CXCR4 to enhance chemotaxis by attachment to chemokine CXCL12, or it can activate TLR4 to create cytokines [50]. HMGB1 interacts with PAMPs, cytokines, and chemokines in addition to its intrinsic activity to increase its extracellular effects [51]. It is interesting to note that IgA- and IgM-mediated immune responses to microorganisms may involve FCAMR. IgA, IgM, and IgG are the main immunoglobulins that prevent bacterial infections from adhering to the mucosal surface in the uterus, as demonstrated by Singh et al. [52].

The *NOS2A* gene, which codes for iNOS, is responsible for the transcriptional regulation of inflammatory mediators generated by immune-competent cells such neutrophils and macrophages [53]. Numerous SNPs within the *NOS2A* gene may have an impact on the generation or activity of iNOS, according to previous research [54].

4.3.2. Role of Examined Metabolic Markers

Mutations in *ADAMTS20* have been linked to endometrial tissue remodeling and inflammation [55]. When compared to un-pregnant pigs throughout the estrous cycle, Kim et al. [56] discovered that *ADAMTS20* was significantly elevated in the endometrium of pregnant pigs during embryo implantation. Furthermore, in beef cattle, *ADAMTS20* was implicated in the postpartum anestrus interval [57].

In first-parity Canadian Holstein cows, *KCNT2* was found to be a potential gene for endometritis 150 days postpartum [58]. Using a genome-wide association analysis (GWAS) and fine mapping investigation for disease features, Freebern et al. [59] recently determined that the *KCNT2* gene is a major candidate for ketosis in dairy cattle. Ketosis and endometritis have a substantial favorable genetic correlation [60]. Therefore, specific loci in *KCNT2* may be promising indicators for next genomic selection procedures to enhance the state of health for both ketosis and metritis.

The targeting of MAPKs that are normally stimulated as a response to inflammatory cytokines and cellular anxiety may be used to treat inflammatory disorders [61]. *FKBP5* was identified by Roy et al. [62] as a potential gene that interacts with poly-chlorinated biphenyls, which are more prevalent in endometritis in humans. A member of the G protein-coupled 7-transmembrane receptor superfamily's leucine-rich repeat-containing subgroup is encoded by gene *RXFP1*. Human placenta accreta and EM were highly correlated with decreased *RXFP1* expression [63]. As it promotes angiogenesis, RXFP1 is crucial for early gestation in horses and other animal species, as noted by Klein [64].

4.3.3. Role of Examined Antioxidant Markers

Endogenous antioxidant indicators, such as CAT and SOD, are examples of the body's own enzymatic and non-enzymatic antioxidant defenses that are implicated in these processes [65]. The initial complex of enzymes in the electron transport chain of the mitochondria, NADH: ubiquinone Oxidoreductase (Complex I), is encoded by the *NDUFS6*

gene [66]. A quantitative trait locus for somatic cell score (SCS) is located in the BTA20 region of the genome, which is home to the *NDUFS6* gene in cattle [67].

HMOX is the rate-limiting enzyme in the heme catabolic pathway, breaking down the heme into equimolar amounts of free iron, biliverdin, and carbon monoxide (CO) [68]. It is also known to be a stress-responsive protein and is thought to have a number of defensive roles in contradiction of different stresses due to its anti-inflammatory, anti-apoptotic, anti-coagulation, anti-proliferative, and vasodilator qualities [69].

### 4.4. Interpretation of the Change in Gene Expression for the Considered Indicators

In sheep, bacterial contamination of the uterine lumen following parturition is frequent [8]. Numerous microbes emerge from the surrounding environment during and after sheep give birth, enter the birth canal, and settle in the uterus [70]. In the ensuing two to four weeks, the majority of healthy sheep are able to naturally remove this pollution [71]. But the formation of clinical endometritis or metritis is dependent on the persistence of pathogenic microorganisms. Polymorphonuclear leukocyte (PMN) phagocytic activity and gene expression are modulated by endocrine and metabolic changes around the time of parturition [72]. Marked elevation of *A2M*, *TLR2*, *IRAK3*, *HMGB1*, *FCAMR*, *iNOS*, *ADAMTS20*, *KCNT2*, *MAP3K4*, *FKBP5*, *RXFP1*, and *HMOX1* was observed in Ossimi ewes with postparturient endometritis. On the other hand, the genes encoding *TGF-β*, *SOD*, *CAT*, and *NDUFS5* were down-regulated. This may be explained by the phagocytic cells' production of proinflammatory cytokines and cytotoxic radicals, both of which seriously inflame the damaged tissue [73]. Furthermore, the overabundance of reactive oxygen species (ROS) resulting from an excess of ROS in the lack of an ideal total antioxidant weakens the immune system [74].

The increased exposure to infections triggers the ewes' immune tissues. The neutrophil recruitment pathway is initiated when lipopolysaccharide (LPS) from Gram-negative bacteria or lipoteichoic acid (LPA) from Gram-positive bacteria is visible to macrophages and epithelial cells. Both of these substances cause the production of TNF and IL1B [75]. Therefore, it is assumed that an infectious agent caused the great mainstream of the ewe-impacted cases of inflammatory reproductive disease in our investigation. We propose that disease resistance can vary due to differences in the potential genes related to these defense tools.

### 5. Conclusions

SNPs for immunological (*A2M*, *TLR2*, *TGF-β*, *IRAK3*, *HMGB1*, *FCAMR*, and *iNOS*), metabolic (*ADAMTS20*, *KCNT2*, *MAP3K4*, *FKBP5*, and *RXFP1*), and antioxidant (*SOD*, *CAT*, *NDUFS5*, and *HMOX1*) indicators were identified in the genes using PCR-DNA sequencing in both normal and endometritis-affected Ossimi ewes. Furthermore, the mRNA amount of these markers varied between ewes with endometritis and those without it. During sheep selection, these exclusive efficient alternatives represent a good prospect to reduce the incidence of endometritis via combining candidate genes with normal welfare. Based on these gene domains, future methods of treating endometritis may be made easier.

**Supplementary Materials:** The following supporting information can be downloaded at: https://www.mdpi.com/article/10.3390/agriculture13122273/s1, Figure S1. *A2M* marker (448 bp) sequences and GenBank gb|XM_012175446.4| for analyzing nitrogenous base alignment for DNA in healthy and endometritis ewes. Figure S2. *TLR2* marker (340 bp) sequences and GenBank gb|DQ890157.1| for analyzing nitrogenous base alignment for DNA in healthy and endometritis ewes. Figure S3. *TGF-β* marker (394 bp) sequences and GenBank gb|NM_001009400.2| for analyzing nitrogenous base alignment for DNA in healthy and endometritis ewes. Figure S4. *IRAK3* marker (431 bp) sequences and GenBank gb|XM_027967477.2| for analyzing nitrogenous base alignment for DNA in healthy and endometritis ewes. Figure S5. *HMGB1* marker (430 bp) sequences and GenBank gb|XM_042254827.1| for analyzing nitrogenous base alignment for DNA in healthy and endometritis ewes. Figure S6. *FCAMR* marker (386 bp) sequences and GenBank gb|XM_042257020.1| for analyzing nitrogenous base alignment for DNA in healthy and endometritis ewes. Figure S7. *ADAMTS20* marker (465 bp)

sequences and GenBank gb|XM_004006435.5| for analyzing nitrogenous base alignment for DNA in healthy and endometritis ewes. Figure S8. *KCNT2* marker (442 bp) sequences and GenBank gb|XM_027976255.2| for analyzing nitrogenous base alignment for DNA in healthy and endometritis ewes. Figure S9. *MAP3K4* marker (398 bp) sequences and GenBank gb|XM_042253698.1| for analyzing nitrogenous base alignment for DNA in healthy and endometritis ewes. Figure S10. *FKBP5* marker (389 bp) sequences and GenBank gb|XM_042237057.1| for analyzing nitrogenous base alignment for DNA in healthy and endometritis ewes. Figure S11. *RXFP1* marker (435 bp) sequences and GenBank gb|XM_012097572.3| for analyzing nitrogenous base alignment for DNA in healthy and endometritis ewes. Figure S12. *SOD* marker (385 bp) sequences and GenBank gb|XR_006060140.1| for analyzing nitrogenous base alignment for DNA in healthy and endometritis ewes. Figure S13. *CAT* marker (397 bp) sequences and GenBank gb|XM_004016396.5| for analyzing nitrogenous base alignment for DNA in healthy and endometritis ewes. Figure S14. *NDUFS5* marker (415 bp) sequences and GenBank gb|XM_027968234.2| for analyzing nitrogenous base alignment for DNA in healthy and endometritis ewes. Figure S15. *HMOX1* marker (377 bp) sequences and GenBank gb|MK630326.1| for analyzing nitrogenous base alignment for DNA in healthy and endometritis ewes. Figure S16. *NOS* marker (360 bp) sequences and GenBank gb|AF223942.1| for analyzing nitrogenous base alignment for DNA in healthy and endometritis ewes.

**Author Contributions:** The PCR was performed, the experiment was developed, and the study report was written by A.A. DNA sequencing and manuscript drafting were assisted by F.A.S. All authors have read and agreed to the published version of the manuscript.

**Funding:** Princess Nourah bint Abdulrahman University Researchers Supporting Project number (PNURSP2023R318); Princess Nourah bint Abdulrahman University, Riyadh, Saudi Arabia also funds publishing of this study.

**Institutional Review Board Statement:** The Mansoura University Animal Care and Use Committee (MU-ACUC), with approval number VM.R.23.10.30, gave its clearance before the animal experiment was carried out.

**Informed Consent Statement:** All farmers provided their informed consent in order to participate in the experiment.

**Data Availability Statement:** The appropriate author will provide supporting information for the study's conclusions upon reasonable request.

**Acknowledgments:** Princess Nourah bint Abdulrahman University Researchers Supporting Project number (PNURSP2023R318), Princess Nourah bint Abdulrahman University, Riyadh, Saudi Arabia acknowledged by the authors. Department of Development of Animal Wealth, Faculty of Veterinary Medicine, Mansoura University, Egypt is also acknowledged by authors.

**Conflicts of Interest:** The authors declare no conflict of interest.

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
