# Peer review of "New Insights into Polymorphisms in Candidate Genes Associated with Incidence of Postparturient Endometritis in Ossimi Sheep (Ovis aries)"

_agriculture, doi:10.3390/agriculture13122273_

Round 1

Reviewer 1 Report

Comments and Suggestions for Authors
Reviewer’s comments on the manuscript by Safhi and Ateya entitled:  New insights on polymorphisms in candidate genes associated with incidence of postparturient endometritis in Ossimi sheep (Ovis arise).

Manuscript ID: agriculture-2687494

November 2023.

This study examined the genes related to immunity, metabolism, and antioxidants that may interact with the prevalence of postpartum endometritis in Ossimi sheep. The manuscript needs to be revised before publication in the Journal.

Specific Comments are as follow:

L99-104: One hundred Ossimi sheep……the examined sheep were divided into three groups of equal size (50 ewes each), this is contradictory.

L116-122: For each gene, please add the full name at the first time, please check the whole manuscript.

L198: Please added amplification efficiency of for the immune, metabolic, and antioxidant genes in Table 2.

L547: A lager number of references, please delete some irrelevant references.

Author Response

Reviewer 1

Comments and Suggestions for Authors

Reviewer’s comments on the manuscript by Safhi and Ateya entitled:  New insights on polymorphisms in candidate genes associated with incidence of postparturient endometritis in Ossimi sheep (Ovis aries).

Manuscript ID: agriculture-2687494

November 2023.

This study examined the genes related to immunity, metabolism, and antioxidants that may interact with the prevalence of postpartum endometritis in Ossimi sheep. The manuscript needs to be revised before publication in the Journal.

Specific Comments are as follow:

Comment

L99-104: One hundred Ossimi sheep……the examined sheep were divided into three groups of equal size (50 ewes each), this is contradictory.

Response

We are grateful to the reviewer for drawing it to our consideration. We apologize for this miswriting type. The sentence is corrected.

Comment

L116-122: For each gene, please add the full name at the first time, please check the whole manuscript.

Response

We are grateful to the reviewer for drawing it to our consideration. The full name is added at first time for each gene at the abstract section then throughout the manuscript, the abbreviation is added.

Comment

L198: Please added amplification efficiency of for the immune, metabolic, and antioxidant genes in Table 2.

Response

We thank reviewer for this. In case of relative expression the amplification efficiency is 100% because we make relative quantification in comparison to the control group.

Comment

L547: A lager number of references, please delete some irrelevant references.

Response

We thank reviewer for this. We have deleted some references without affecting the quality of the manuscript.

Reviewer 2 Report

Comments and Suggestions for Authors

The paper describes associations of polymorphisms in candidate genes with incidence of endometritis p.p. within a regional sheep breed. Needless to say that there is a growing interest to understand the interaction between polymorphism in genes, to identify candidate genes and to use them for improving metabolic, reproductive or immune functions. Furthermore, it is important to support local breeds to strengthen the commitment for sustainability.

Both, the Introduction and Results have been shortened considerably, and the Discussion has also been edited to remove some of the less important material. Please define clearly the objectives of this study (L92/93 is confusing).

My major concerns with the paper are:

1. The integrity of DNA and RNA is required and must be maintained after blood collection and during storage. How did you manage it under expected high temperatures in this exposed region?

2. Were the authors happy with the performance of the kits SensiFastTM SYBR, Bioline, CAT No. Bio98002 and the SYBR Green PCR Master Mix (Toronto, Ontario, Canada: Quantitect SYBR green PCR reagent, Catalogue No. 204141)? No control data are provided.

3. What about the third group? L102 is confusing.

4. How would you interpret the divergence in the body weight of the ewes (L 40 vs. L 100).

5. Did you not expect differences in expression of metabolic and immune genes between dams of firstborns and multipara? The average age of the ewes is 3.5 years. The metabolic and immune related results should be distinguished between nulliparous and multiparous ewes.

6. The authors did not access whether the affected ewes (as described in  L107/108  with very prominent clinical symptoms) had medical cured.  If not:  Than we have a conflict to the ethical statement of the Mansoura University Animal Care and Use Committee and the manuscript should be declined. If yes: No treatment was mentioned (!) and the reviewer argues that there is an effect of the medical treatment on the results? Furthermore, the reviewer argues that there is a huge variation of the interval between lambing and blood collection. No exact data regarding blood collection and the processing of the samples are provided. So, the time relative to disease onset and sampling is difficult to interpret.

7. How would you discuss the different metabolic results in relation to different milk performance (yield and day in milk - not shown in the paper)?

8. The authors compared healthy ewes and dams with severe symptoms of endometritis. How did you exclude ewes with subclinical endometritis from your study? Do you not expect an influence on your metabolic and immune results using of ewes with subclinical endometritis?

Some minor concerns:

L279/280 are confusing and not possible to follow

 Some English terms and medical phrases are uncommon and misleading (poke, coloured discharge, each vein etc.)

Comments on the Quality of English Language

see above

Author Response

Reviewer 2

Comments and Suggestions for Authors

The paper describes associations of polymorphisms in candidate genes with incidence of endometritis p.p. within a regional sheep breed. Needless to say that there is a growing interest to understand the interaction between polymorphism in genes, to identify candidate genes and to use them for improving metabolic, reproductive or immune functions. Furthermore, it is important to support local breeds to strengthen the commitment for sustainability.

Comment

Both, the Introduction and Results have been shortened considerably, and the Discussion has also been edited to remove some of the less important material. Please define clearly the objectives of this study (L92/93 is confusing).

Response

  • We thank reviewer for this. The objectives of the study is clearly defined.
  • Concerning introduction we have deleted some sentences have the same meaning so the introduction is changed to contain the following items (importance of sheep, information on the studied breed, endometritis and its complications and possible treatments based on molecular approaches, importance of molecular biology in choosing the candidate genes for certain trait, and the objectives of the study).
  • Discussion section had been considerably shortened to contain the following items (Discussion for gene polymorphism results, discussion for gene expression results, discussion for the role of each chosen marker in reproduction and deciphering for selection of these markers in our investigation, and interpretation for changes in the gene expression results in the studied markers).
  • Results section contains full description for gene polymorphisms results especially the discovered SNPs are reported here for first time alongside the effect of mutation in each gene (Synonymous or non-synonymous). The second part of results contain gene expression results. Noteworthy mentioning that our study is the first that reports the possible association between the investigated immune, metabolic, and antioxidant markers in sheep.

My major concerns with the paper are:

Comment

  1. The integrity of DNA and RNA is required and must be maintained after blood collection and during storage. How did you manage it under expected high temperatures in this exposed region?

Response

  • We thank reviewer for this. Freshly collected blood samples were sent to lab without delay (At the same day) to avoid degradation of DNA, and RNA. During collection of blood samples, we make our precautions by taking ice box to the site of collection. Each collected blood sample is put without delay into ice pox. Finally all samples are confirmed to be covered with ice at the site of collection and during transportation to lab.
  • We have added a sentence indicating this issue.

Comment

  1. Were the authors happy with the performance of the kits SensiFastTM SYBR, Bio‐line, CAT No. Bio‐98002 and the SYBR Green PCR Master Mix (Toronto, Ontario, Canada: Quantitect SYBR green PCR reagent, Catalogue No. 204141)? No control data are provided.

Response

  • We thank reviewer for this. The gene expression approach includes three steps (RNA extraction, cDNA synthesis, and Real time PCR).
  • Actually we know that the extraction of RNA from blood is somewhat difficult; therefore we have applied more than one kits until we achieved the best results with this kit.
  • We have done more than one published research using these kits. Please see the following publications:
  • Al-Sharif, M.; Ateya, A. New Insights on Coding Mutations and mRNA Levels of Candidate Genes Associated with Diarrhea Susceptibility in Baladi Goat. Agriculture 2023, 13, 143. https://doi.org/10.3390/ agriculture13010143.
  • Ateya, A.; Safhi, F.A.; El-Emam, H.; Marawan, M.A.; Fayed, H.; Kadah, A.; Mamdouh, M.; Hizam, M.M.; Al-Ghadi, M.Q.; Abdo, M.; et al. Combining Nucleotide Sequence Variants and Transcript Levels of Immune and Antioxidant Markers for Selection and Improvement of Mastitis Resistance in Dromedary Camels. Agriculture 2023, 13, 1909. https://doi.org/ 10.3390/agriculture13101909.
  • Ateya A, Al-Sharif M, Abdo M, Fericean L, Essa B. Individual Genomic Loci and mRNA Levels of Immune Biomarkers Associated with Pneumonia Susceptibility in Baladi Goats. Vet Sci. 2023 Mar 1;10(3):185. doi: 10.3390/vetsci10030185. PMID: 36977224; PMCID: PMC10051579.
  • Essa B, Al-Sharif M, Abdo M, Fericean L, Ateya A. New Insights on Nucleotide Sequence Variants and mRNA Levels of Candidate Genes Assessing Resistance/Susceptibility to Mastitis in Holstein and Montbéliarde Dairy Cows. Vet Sci. 2023 Jan 3;10(1):35. doi: 10.3390/vetsci10010035. PMID: 36669036; PMCID: PMC9861242.
  • Darwish, A.; Ebissy, E.; Ateya, A.; El-Sayed, A. Single nucleotide polymorphisms, gene expression and serum profile of immune and antioxidant markers associated with postpartum disorders susceptibility in Barki sheep. Anim. Biotechnol. 2023, 34, 327-339. 

Comment

  1. What about the third group? L102 is confusing.

Response

We are grateful to the reviewer for drawing it to our consideration. We apologize for this miswriting type. The sentence is corrected.

Comment

  1. How would you interpret the divergence in the body weight of the ewes (L 40 vs. L 100).

Response

  • We thank reviewer for this. In line 40, this part in introduction section. In this part we talk about the Ossimi sheep in general. Line 100; this part in materials and methods section. In this part we talk about the actual used Ossimi sheep in our investigation i.e. specific animals used in our study.

Comment

  1. Did you not expect differences in expression of metabolic and immune genes between dams of firstborns and multipara? The average age of the ewes is 3.5 years. The metabolic and immune related results should be distinguished between nulliparous and multiparous ewes.

Response

  • We thank reviewer for this. The investigated ewes were from the same group and litter size. In addition the investigated ewes were nearly the same age and share the same environmental conditions.
  • On genetic bases we make fixation to the environmental factors to make an accurate judging on phenotypic variations based on genotype.
  1. The authors did not access whether the affected ewes (as described in L107/108 with very prominent clinical symptoms) had medical cured.  If not:  Than we have a conflict to the ethical statement of the Mansoura University Animal Care and Use Committee and the manuscript should be declined. If yes: No treatment was mentioned (!) and the reviewer argues that there is an effect of the medical treatment on the results? Furthermore, the reviewer argues that there is a huge variation of the interval between lambing and blood collection. No exact data regarding blood collection and the processing of the samples are provided. So, the time relative to disease onset and sampling is difficult to interpret.

Response

  • We thank reviewer for this. In ranked scientific journals a signed consent form from the owner is provided with the manuscript.
  • The main purpose of our study is differentiating between two groups healthy and endometritis affected ewes. So we collected blood samples from the two groups. On basis of our investigation, there is no necessity for taking the third group. Based on farm records, once the disease appear we take blood samples from the affected animals. We track the healthy animals over a long period to guarantee that it is in good condition. Once it is ascertained, we take also a blood sample. However, in fact the diseased animals are treated by experienced veterinarians under observations and care of experienced farmers.
  • We have published previous works shared the same concept on endometritis as follows:
  • Darwish, A.; Ebissy, E.; Ateya, A.; El-Sayed, A. Single nucleotide polymorphisms, gene expression and serum profile of immune and antioxidant markers associated with postpartum disorders susceptibility in Barki sheep. Anim. Biotechnol. 2023, 34, 327-339. 
  • Al-Sharif, M.; Abdo, M.; Shabrawy, O.E.; El-Naga, E.M.A.; Fericean, L.; Banatean-Dunea, I.;    Ateya, A. Investigating    Polymorphisms and Expression    Profile of Immune, Antioxidant, and Erythritol-Related Genes for    Limiting Postparturient    Endometritis in Holstein Cattle. Vet. Sci. 2023, 10, 370. https://doi.org/ 10.3390/vetsci10060370.
  • Mona Al-Sharif, Basma H. Marghani & Ahmed Ateya (2023) DNA polymorphisms and expression profile of immune and antioxidant genes as biomarkers for reproductive disorders tolerance/susceptibility in Baladi goat, Animal Biotechnology, 34:7, 2219-2230, DOI: 10.1080/10495398.2022.2082975.
  • According to previous publications, we taken in our considerations the medical treatment does not influence the results because once the disease appear we collect blood sample so there is no influence of treatments on results.
  • Precautions during blood sampling till laboratory analysis are added upon your inquiry.
  • Concerning there is a huge variation of the interval between lambing and blood collection, we collect blood samples from all animals at the beginning at the the same time and we track all animals all over the transition period to be judged it is healthy or diseased

Comment

  1. How would you discuss the different metabolic results in relation to different milk performance (yield and day in milk - not shown in the paper)?

Response

  • We thank reviewer for this. The main purpose of our study is to differentiate between the healthy and endometritis affected ewes on genetic basis.
  • We have submitted a preliminary abstract before sending the whole manuscript to the journal. After that the abstract is accepted by the academic and website editors.
  • Our paper contain the following potential unique points:
  • The immunological alterations and genetic variants associated with postpartum endometritis in sheeps are equally little understood, and few reliable diagnostic and prognostic instruments are available to help us in the development of our preventive and treatment strategies.
  • It is crucial to stress that the differences found and the materials easily accessible in this context provide new insights into the markers under investigation when matched to the relatable sequence acquired from GenBank.
  • Employing genome-wide association analysis for our investigated markers, recent studies have identified distinct genes that are specifically linked to the prevalence of bovine endometritis. Nevertheless, no investigation has yet examined the connection between endometritis risk and the SNPs in these genes.
  • As far as we are aware, no previous research has examined the relationship between postparturient endometritis in Ossimi sheeps and the variation of immunological (A2M, TLR2, TGF-β, IRAK3, HMGB1, FCAMR, and iNOS), metabolic (ADAMTS20, KCNT2, MAP3K4, FKBP5, and RXFP1) and antioxidant (SOD, CAT, NDUFS5, and HMOX1) markers.
  • The mRNA levels of these markers and their correlation with the incidence of postpartum endometritis in sheeps are being examined for the first time in this investigation. We performed SNP genetic markers and gene expression to analyze genetic variation, overcoming the limitations of previous investigations. As a result, the tools regulating the examined genes are well assumed by both normal and endometritis ewes.
  • Depending on the respected reviewer, we will focus on the reviewer inquiry in further investigations

Comment

  1. The authors compared healthy ewes and dams with severe symptoms of endometritis. How did you exclude ewes with subclinical endometritis from your study? Do you not expect an influence on your metabolic and immune results using of ewes with subclinical endometritis?

Response

  • We thank reviewer for this. We collected blood samples from the clinical cases. We added the word clinical to materials and methods section.
  • To say the animal is healthy, we tracked the animal for the transition period to guarantee that it is in good wellbeing.
  • We have published more than one work on endometritis in different livestock depending on general health condition of animal as well as the characters of the vaginal discharge as follows:
  • Darwish, A.; Ebissy, E.; Ateya, A.; El-Sayed, A. Single nucleotide polymorphisms, gene expression and serum profile of immune and antioxidant markers associated with postpartum disorders susceptibility in Barki sheep. Anim. Biotechnol. 2023, 34, 327-339. 
  • Al-Sharif, M.; Abdo, M.; Shabrawy, O.E.; El-Naga, E.M.A.; Fericean, L.; Banatean-Dunea, I.;    Ateya, A. Investigating    Polymorphisms and Expression    Profile of Immune, Antioxidant, and Erythritol-Related Genes for    Limiting Postparturient    Endometritis in Holstein Cattle. Vet. Sci. 2023, 10, 370. https://doi.org/ 10.3390/vetsci10060370.
  • Mona Al-Sharif, Basma H. Marghani & Ahmed Ateya (2023) DNA polymorphisms and expression profile of immune and antioxidant genes as biomarkers for reproductive disorders tolerance/susceptibility in Baladi goat, Animal Biotechnology, 34:7, 2219-2230, DOI: 10.1080/10495398.2022.2082975.
  • The main purpose of our study is using the candidate gene approach for monitoring the health status of endometritis affected ewes to offer a new approach for treatment and selection of resistant animal otherwise the animal suffer from clinical or subclinical condition. This point coincides with the main scope of the special issue of the journal (Welfare, Behavior and Health of Farm Animals).

Some minor concerns:

  • Comment

L279/280 are confusing and not possible to follow

Response

We are grateful to the reviewer for drawing it to our consideration. It is correted.

  • Comment

  Some English terms and medical phrases are uncommon and misleading (poke, coloured discharge, each vein etc.)

Response

We are grateful to the reviewer for drawing it to our consideration. The manuscript is revised and English edited.

Reviewer 3 Report

Comments and Suggestions for Authors

The authors presented a study of genes associated with immune-related, metabolic, and antioxidant genes that affect endometritis. It was demonstrated experimentally that these marker nucleotide variants and gene expression patterns can predict endometritis incidence. These findings provide a new approach for the future treatment of endometritis. This study has certain clinical significance, but there are some issues that need to be revised and improved. Specifically, the review comments are as follows:

1. It is suggested that the summary section be streamlined by deleting the phrase "To extract DNA and RNA, the blood was drawn into tubes containing EDTA anticoagulant and vacuum‐sealed to obtain whole blood".

2. It is suggested to add "nucleotide variation" or other closely related terms in the keywords, so as to describe the article more comprehensively. This will more fully describe the topic of the article. 

3. whether the ewes used in this study were from the same group and litter size. 4. whether the ewes selected for immunization at the beginning of this study were from the same group and litter size.

4. what were the bases for selecting immune-related, metabolic, and antioxidant-related genes at the beginning of this study?

5. add an "s" to the word "sheep" in the text, and note the corrections throughout the text.

6. Please provide a copy of the original gel electrophoresis from the experimental study.

Comments on the Quality of English Language

The authors have some ability to write scientific and technical papers in English, and write logically and appropriately, with no serious descriptive problems in English.

Author Response

Dear respected reviewer thank you very much for your effort in the revision of the manuscript. Attached is our revision. we would it will be satisfactory.

Round 2

Reviewer 1 Report

Comments and Suggestions for Authors

I have gone through the revised manuscript, and I am satisfied with the changes made by the authors. 

Author Response

Dear reviewer

Thank you very much for your kind response and satisfaction with our changes to manuscript

Reviewer 2 Report

Comments and Suggestions for Authors

The authors have much improved their ms “New insights on polymorphisms in candidate genes associated with incidence of postparturient endometritis in Ossimi sheep”. All chapters, which grabbed the attention of the reviewer some weeks ago, is to undergo a thorough makeover. Many hints and were incorporated and raised questions have been picked up. The response regarding Q.5 is very poor and unsatisfactory; especially because the physiological background is neglected. The reviewer is happy that all obligation and principles regarding animal welfare have been observed. However, the declaration that a performed efficient treatment has no effect is definitely overstated and should be rethought. It should be clearly stated that the blood samples were taken before the immediate medical treatment.

Author Response

Comment

The authors have much improved their ms “New insights on polymorphisms in candidate genes associated with incidence of postparturient endometritis in Ossimi sheep”. All chapters, which grabbed the attention of the reviewer some weeks ago, is to undergo a thorough makeover. Many hints and were incorporated and raised questions have been picked up. The response regarding Q.5 is very poor and unsatisfactory; especially because the physiological background is neglected. The reviewer is happy that all obligation and principles regarding animal welfare have been observed. However, the declaration that a performed efficient treatment has no effect is definitely overstated and should be rethought. It should be clearly stated that the blood samples were taken before the immediate medical treatment.

Response

We thank reviewer for this.  The following sentence is added the blood samples were taken before the immediate medical treatment upon your inquiry.